# E-DETR: Evidential Deep Learning for End-to-End Uncertainty Estimation in Object Detection

## Abstract

Detection transformers (DETR) have emerged as powerful end-to-end learning frameworks for object detection, directly regressing detection parameters as point estimates. However, these networks often lack the ability to express any uncertainty within their estimates. In this work, we replace the regression of point estimates with the direct learning of the posterior distribution in a sampling-free manner by leveraging deep evidential learning, complementing the end-to-end DETR architecture. We present an instance-aware uncertainty framework by extending evidential deep learning with an IoU-aware loss, jointly modelling both classification and localization uncertainties. Furthermore, we enable the model to leverage its uncertainty for self-calibration, aligning the predicted probabilities with the true likelihood of outcomes, and effectively apply evidential deep learning for the task of imbalanced dense object detection. Our approach is easily extensible and requires only fine-tuning, thus leveraging the pre-training of transformers on large datasets. We conduct extensive experiments on two in-domain and three out-of-domain datasets, demonstrating impressive improvements in generalization performance, especially when fine-tuning on heavily imbalanced datasets characterized by data scarcity.

## 1 Introduction

Object detection is a fundamental aspect of machine vision, including in safety-critical applications such as perception in autonomous vehicles (Balasubramaniam & Pasricha, 2022). While detection accuracy is crucial, it is equally important to quantify the uncertainty associated with the detection estimates. However, state-of-the-art object detection networks predominantly rely on deep neural networks, which are often viewed as black-box models (Liang et al., 2021). This presents a significant challenge as it becomes difficult to assess the trustworthiness of these networks and to understand their decision-making processes. Despite notable advancements in deep learning and computer vision over the past decade, there remains a significant gap in research in understanding and quantifying the uncertainty associated with the outputs of such object detection networks.

Addressing uncertainty in object detection is particularly challenging because it must account for both localization and classification uncertainties (Feng et al., 2021). Localization uncertainty involves the accuracy of predicted bounding boxes, which can be affected by factors such as occlusions and complex backgrounds. Meanwhile, classification uncertainty pertains to the confidence in the predicted class labels, often influenced by ambiguous visual features and out-of-distribution data. The need to jointly model these uncertainties complicates the development of effective detection systems, as traditional methods often treat them separately, as a multi-task problem (Oksuz et al., 2020). This separation can hinder the ability to provide a comprehensive understanding of the overall uncertainty associated with each detection.

Previous studies addressing uncertainty in object detection for autonomous vehicles have typically relied on sampling-based methods, such as Bayesian neural networks and Monte Carlo dropout (Gawlikowski et al., 2023). While these methods are effective in quantifying uncertainty, they are computationally expensive, require multiple runs to sample different results, and often result in reduced overall accuracy due to factors such as biased sampling, improper weighting, and the removal

of parameters caused by dropout in Monte Carlo methods (Feng et al., 2021). Ever since the emergence of evidential deep learning (Sensoy et al., 2018; Amini et al., 2020), there has been increased interest in leveraging evidence theory for object detection (Park et al., 2023; Nallapareddy et al., 2023), estimating uncertainty by predicting prior distributions of posterior models in a sampling-free manner. However, the existing object detection networks required various post-processing techniques to derive the final object estimates, and incorporating evidential learning necessitated sophisticated modifications to the base detection networks requiring significant pre-training.

Moreover, evidential deep learning is not without its challenges, particularly concerning biases that can arise in the context of object detection. One significant issue is class imbalance (Xia et al., 2022), which is prevalent in real-world driving scenarios due to the varying densities of objects in each frame. When certain object classes are observed significantly more frequently than others, the evidential model may develop a bias towards these dominant classes. This bias can lead to overconfidence in predictions for the more common classes while underestimating uncertainty for less frequent classes, resulting in poor performance in detecting under-sampled objects.

Another approach is to focus on model calibration (Pathiraja et al., 2023; Kuzucu et al., 2024; Munir et al., 2024) which aims to align the predicted class probabilities and bounding box localization with the true likelihood of outcomes. By refining both the confidence scores and the spatial accuracy of bounding boxes produced by object detection models, calibration techniques can also provide more reliable estimates without the need for multiple forward passes through the network.

Recently, Carion et al. (2020) demonstrated that detection transformers (DETR) can be effectively used for end-to-end object detection, by framing it as a sequence-to-sequence problem, regressing detection estimates as an unordered sequence *i.e.* as a set prediction problem. Their approach eliminates the need for hand-crafted components such as Non-Maximum Suppression (NMS) and anchor formulation. In this paper, we extend DETR with evidential deep learning to incorporate uncertainty estimation, enabling the direct output of detection estimates and their associated uncertainties in an end-to-end, sampling-free manner.

## 1.1 PROBLEM FORMULATION AND CONTRIBUTIONS

In this work, we address multiple challenges:

1. Evidential deep learning is typically ill-posed for object detection due to the significant class imbalance between foreground and background objects, as well as their varying densities from frame to frame. In this work, we address class imbalance and adapt evidential deep learning for object detection. Furthermore, we leverage model uncertainty during training to implicitly self-calibrate model confidence to align with the true likelihood of outcomes, using the model calibration error to modulate uncertainty regularization.

2. Existing methods address localization and classification uncertainty in object detection as independent tasks within a multi-task framework. We present an instance-aware uncertainty framework by extending evidential deep learning with an IoU-aware loss, enabling the joint modelling of both localization and classification uncertainties.

3. Previous works in the literature that incorporated evidential deep learning in object detection required sophisticated transforms to their base object detection networks. In this paper, we propose a simple, easily extensible, end-to-end approach for integrating evidential learning into object detection using DETR. To the best of our knowledge, this represents the first adaptation of DETR to incorporate evidence theory within its learning framework for supervised learning. Our approach is easily extensible and can be applied to any DETR architecture. Our method does not require any pre-training, *i.e.* it can be implemented using only fine-tuning.

4. DETR supervises one-to-one matching and utilizes a non-differentiable bipartite graph matching for ranking object queries, which can lead to training instability due to stochastic optimization. We incorporate uncertainty estimates during the matching process, providing a probabilistic framework for informed decision-making in the assignment of object queries to ground truth instances. This approach not only improves generalization but also reduces the model variations observed across multiple training sessions.

## 2 BACKGROUND AND RELATED WORK

Prior research employing evidential deep learning in the context of supervised object detection is quite limited. This includes Park et al. (2023), who argued that evidential deep learning (Sensoy et al., 2018) couldn't be directly applied to traditional object detection networks such as Faster R-CNN (Girshick, 2015) and SSD (Liu et al., 2016). They noted that learning the parameters of a posterior Dirichlet distribution (Jsang, 2018) for dense object detection, as opposed to simple classification, lead to training instability and overly uncertain predictions. To address the limitations associated with learning a Dirichlet distribution as originally proposed by Sensoy et al. (2018), they augmented the learning framework and introduced a new Model Evidence Head module to effectively harness evidential learning for object detection. However, networks like Faster R-CNN and SSD still require significant overhead, including anchor formulation and non-maximum suppression (NMS).

Additionally, Nallapareddy et al. (2023) proposed an extension to CenterNet (Zhou et al., 2019) and modelled uncertainty as Gaussian heatmaps utilizing 3D convolutional layers to estimate objectness uncertainty as a binary value. However, their method was inherently conditioned on the compute-intensive operation of estimating high-resolution heatmaps, and required additional regularization parameters for accurate localization of object centeredness. While CenterNet is more efficient compared to traditional detection models, it still necessitates post-processing to derive final predictions from Gaussian heatmaps.

In contrast, our approach, based on the DETR framework (Carion et al., 2020), simplifies the training process by eliminating the need for anchor boxes and NMS, which are common complications in traditional object detection methods. Traditional models often struggle with the design and tuning of anchor boxes, which can lead to suboptimal performance if not carefully calibrated. Furthermore, the reliance on post-processing techniques like NMS can introduce additional computational overhead and potential errors in object localization. Our method, being highly generalizable, can be seamlessly applied to any DETR model.

## 3 UNCERTAINTY ESTIMATION

Typically, neural networks are trained to generate predictions by minimizing a loss function that quantifies an error between the predicted outcomes and the actual results. However, these networks often lack a mechanism to express the level of confidence associated with their predictions. Moreover, in object detection, biases such as class imbalance can hinder a model's ability to learn accurate representations of real-world scenarios, as underrepresented classes may not be adequately captured during training, leading to biased predictions and a failure to generalize effectively.

To address these challenges, we employ evidential deep learning in conjunction with an instance-aware class balancing loss. Evidential deep learning (Sensoy et al., 2018; Amini et al., 2020) is a framework that extends traditional neural networks by modelling uncertainty in predictions through the use of evidence theory or the Dempster–Shafer Theory of Evidence (Dempster, 1968).

### 3.1 INSTANCE AWARE UNCERTAINTY ESTIMATION IN OBJECT CLASSIFICATION

In a classification task, the final layer of a neural network often uses an activation function like softmax. The softmax function converts the raw output scores (logits) from the previous layer into probabilities that sum to 1. This means that for a multi-class classification problem with $K$ classes, the output can be represented as:

$$p_k = \frac{e^{z_k}}{\sum_{j=1}^{K} e^{z_j}}, \tag{1}$$

where $z_k$ are the logits for class $k$.

The softmax function provides a point estimate for class probabilities but does not inherently quantify uncertainty. For instance, if the softmax outputs probabilities of $[0.7, 0.2, 0.1]$, it indicates a strong belief in the first class, but it does not express how confident the model is in that estimate or how much it might vary with noisier inputs.

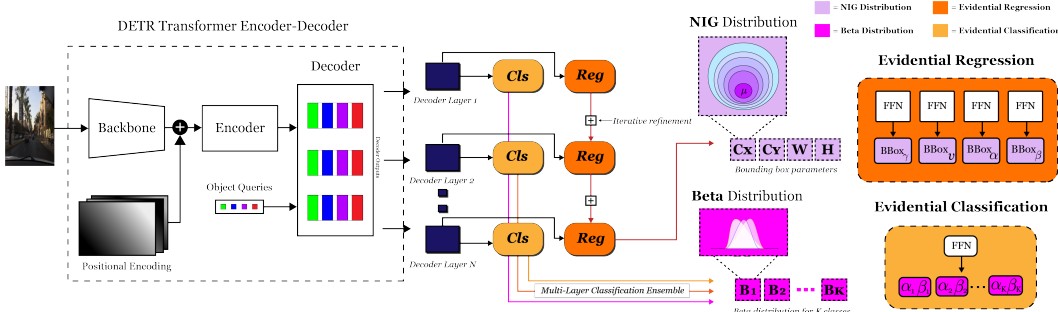

Figure 1: Network architecture: We replace the conventional approach of learning point estimates with the direct estimation of posterior distributions. For bounding boxes, we learn the Normal Inverse Gamma (`NIG`) distribution for each parameter. Each decoder layer regresses estimates for the target bounding box parameters sampling from a common distribution, the distribution parameters are iteratively refined every layer. For classification, each decoder layer learns a Beta distribution for all classes. We aggregate the outputs from all layers to quantify both aleatoric and epistemic uncertainties.

To address this limitation, we aim to represent class confidence as a probability distribution rather than as a single-point estimate. Typically, this is achieved by capturing the posterior distribution through sampling from multiple runs. Utilizing the posterior distribution allows us to reflect the likelihood of various outcomes and to better understand the uncertainty associated with predictions. However, Sensoy et al. (2018) demonstrated that deep neural networks can directly estimate the posterior distribution in a sampling-free manner by learning the parameters of a Dirichlet distribution (Jsang, 2018), providing an end-to-end approach eliminating the redundant computational overhead.

The Dirichlet distribution is parameterized by a vector of positive reals, often referred to as pseudo-counts, denoted as $\boldsymbol{\alpha} = [\alpha_1, \alpha_2, \ldots, \alpha_K]$, where $K$ is the number of classes and $\alpha_k$ represents the belief for the $k^{th}$ class. The probability density function of the Dirichlet distribution is given by:

$$D(\mathbf{p}|\boldsymbol{\alpha}) = \frac{1}{B(\boldsymbol{\alpha})} \prod_{k=1}^{K} p_k^{\alpha_k - 1},$$

(2)

where $\mathbf{p} = [p_1, p_2, \ldots, p_K]$ is a vector of probabilities such that $p_k \geq 0$ and $\sum_{k=1}^{K} p_k = 1$. Here, $B(\boldsymbol{\alpha})$ is the normalization constant, known as the $K$-dimensional multinomial beta function (Kotz et al., 2019).

However, the number of samples, *i.e.*, the frequency per class, can significantly impact the estimation of parameters for the Dirichlet distribution. Class imbalance can substantially affect the performance of models, leading to an overestimation of the likelihood for the majority class while underestimating that of minority classes (Cui et al., 2019). This issue is particularly pronounced in object detection scenarios, where class imbalance is challenging to address due to the varying densities of objects present in each frame.

To address the challenges posed by class imbalance, RetinaNet (Lin et al., 2017) introduced the focal loss for Faster R-CNN, specifically designed for dense object detection. The focal loss is an adaptation of the standard cross-entropy loss that adds a modulating factor to the loss function, and for a binary class can be represented as:

$$\text{FL}(\text{p}, \text{y}) = \begin{cases} -\kappa(1-\text{p})^{\gamma}\log(\text{p}) & \text{if} \quad \text{y} = 1 \\ -(1-\kappa)\text{p}^{\gamma}\log(1-\text{p}) & \text{otherwise,} \end{cases}$$

(3)

where $\text{y} \in [0, 1]$ denotes the class label, $\text{p} \in [0, 1]$ the predicted class probability, $\kappa$ a balancing factor to address class imbalance, and $\gamma$ a focusing parameter that adjusts the rate at which easy examples are down-weighted. A higher value of $\gamma$ puts more focus on hard-to-classify examples.

As explored by Mukhoti et al. (2020), the focal loss effectively self-calibrates the learned model by minimizing a regularized Kullback-Leibler (KL) divergence between the predicted and target distributions. This approach not only reduces the KL divergence but also enhances the entropy of

the predicted distribution, thereby mitigating the risk of the model becoming overconfident in its predictions.

Building on this, VFNet (Zhang et al., 2021) introduced the varifocal loss, which also incorporated localization by integrating an Intersection over Union (IoU)-aware Classification Score (IACS). The score could serve as a joint representation of both object presence confidence and localization accuracy, further refining the model's ability to handle class imbalance in dense object detection scenarios. The varifocal loss can be defined as:

$$\text{VFL}(p, q) = \begin{cases} -q(q\log(p) + (1 - q)\log(1 - p)) & q > 0 \\ -\kappa p^{\gamma}\log(1 - p) & q = 0, \end{cases} \tag{4}$$

where $p$ is the predicted IACS score and $q$ denotes the target score. The value of $q$ is determined by the IoU between the predicted bounding box and the ground truth.

**Instance-aware uncertainty estimation**  In light of these insights, we present a new instance-aware uncertainty estimation scheme. Instead of employing a multivariate Dirichlet distribution, we learn the parameters of a univariate Beta distribution (Forbes et al., 2011) and individually model and modulate uncertainty for each class, and overcome class imbalance.

The Beta distribution is characterized by two positive parameters, $\alpha$ and $\beta$. Here, $\alpha$ represents the evidence supporting that an object belongs to the class, while $\beta$ reflects the evidence indicating that the object is not of that class, and is part of the background respective to the class. The problem can also be framed as learning the 'objectness' of feature representations for that class. For each class, we gather evidence of whether the detected feature representation is an object or not an object. From Equation (2), the probability density function of the Beta distribution for a binary class can be defined as:

$$\mathbf{B}(p|\alpha, \beta) = \frac{p^{\alpha - 1}(1 - p)^{\beta - 1}}{B(\alpha, \beta)}. \tag{5}$$

The probability $p$ and uncertainty $u$ for a binary class can be computed as:

$$p = \frac{\alpha}{\alpha + \beta}, \qquad u = \frac{2}{\alpha + \beta}. \tag{6}$$

We also introduce a few changes to the sum of squares loss proposed by (Sensoy et al., 2018) for uncertainty estimation. Unlike Sensoy et al. (2018), we apply an exponent rather than the ReLU function to derive our evidence, as empirically it performs better. Given a sample $i$, let $f(\mathbf{x}_i|\Theta)$ denote the evidence vector predicted by the network for classification, where $\Theta$ represents the network parameters. As illustrated in Fig 1, for each class $k$, the network outputs parameters $\alpha_{ik}$ and $\beta_{ik}$ for its corresponding Beta distribution, where $\alpha_{ik} = \exp(z_{\alpha k}) + 1$, $\beta_{ik} = \exp(z_{\beta k}) + 1$, and $z_{\alpha k}$, $z_{\beta k}$ are the output logits. Let $\mathbf{y}_i$ be a one-hot vector encoding the ground-truth class of observation $\mathbf{x}_i$ and $y_{ik} = 0$ for all $k \neq j$. We can plug in the parameters into the sum of squares evidential classification loss proposed by (Sensoy et al., 2018) where:

$$\mathcal{L}_i(\Theta) = \sum_{j=1}^{K}(y_{ij} - p_{ij})^2 + \frac{p_{ij}(1 - p_{ij})}{\alpha_{ij} + \beta_{ij} + 1}. \tag{7}$$

Then, we introduce IoU scores to integrate localization accuracy and compute instance-aware class weights, which can be applied as:

$$\mathcal{W}_{IoU_i} = \sum_{j=1}^{K} \kappa_j p_{ij}^{\gamma_j}(1 - y_{ij}) + q_{ij}, \tag{8}$$

where $\kappa_j$ is the class balancing factor, $\gamma_j$ class-specific focusing parameter, $y_{ij}$ a binary label, and $q_{ij}$ the IoU between the predicted bounding box and the ground truth. The class confidence is down-weighted for samples with a lower IoU score.

**"I don't know" regularization**  A Beta distribution with no evidence is equivalent to a uniform distribution, implying all outcomes are equally unlikely. To maintain predictions in line with this

notion of uncertainty, characterized by "I don't know," Sensoy et al. (2018) added an adversarial KL divergence term to the loss function. This term ensures that predictions remain conservative and do not become overly confident in situations where there is insufficient information about the outcomes.

However, in the context of object detection, the adversarial regularization hinders learning and causes training instability, as the background uncertain predictions heavily outnumber the limited instances of foreground classes. For object classification, Sensoy et al. (2018) proposed an annealing loss to prevent premature convergence to the uniform distribution for misclassified samples, which could potentially be classified correctly in later epochs.

**Uncertainty aware self-calibration** To alleviate the adversarial interaction, we let the model be self-aware of its uncertainty and introduce self-calibration regularization. Our method not only stabilizes training but implicitly calibrates the model. The evidence is dynamically weighed based on the confidence of the predictions. When the predicted probability deviates from the actual outcome, the regularization term is increased, thereby reducing the accumulated evidence and forcing the distribution towards a state of maximum uncertainty. Conversely, when the predicted probability aligns closely with the actual outcome, the regularization term is decreased, allowing the model to accumulate more evidence, and reinforcing the learned distribution. This approach ensures that the model remains adaptable, penalizing overconfident predictions that are incorrect while encouraging exploration of the parameter space when uncertainty is high.

We extend the standard evidential loss by incorporating the model calibration error to dynamically modulate uncertainty regularization. The loss is then weighted with the IoU score for instance-aware uncertainty estimates. The overall calibrated evidential classification loss function can be defined as:

$$\mathcal{L}_{class}(\Theta) = \left( \mathcal{L}(\Theta) + \lambda_{ann} \underbrace{|y - p|}_{\text{Model Calibration Error}} \mathcal{L}_{KL} \right) \odot \underbrace{\mathcal{W}_{IoU}}_{\text{Instance-Aware Weight}}, \tag{9}$$

where $\mathcal{L}_{KL}$ is the adversarial KL divergence loss and $\lambda_{ann}$ is the annealing regularization coefficient.

## 3.2 UNCERTAINTY AWARE OBJECT LOCALISATION

In a typical bounding box regression task, the objective is to learn a function $f(\mathrm{x}; \Theta)$ that can accurately estimate the average correct output for a given input. This is usually achieved by minimizing the mean squared error loss, defined as:

$$\mathcal{L}_i(\Theta) = \frac{1}{2}||y_i - f(x_i; \Theta)||^2. \tag{10}$$

While this formulation effectively trains the model to approximate the expected bounding box coordinates, it does not account for its prediction uncertainty. Specifically, it lacks any explicit representation of the underlying noise or variability in the data during the estimation process.

Amini et al. (2020) explain how the Dirichlet distribution can be further exploited for the task of learning the epistemic and aleatoric uncertainty in regression. The Dirichlet distribution is typically well-suited for modelling uncertainty in scenarios with categorical outcomes. However, by treating multiple detections of the same object as samples from a distribution, we can leverage the Dirichlet distribution to capture the variability in localisation predictions.

We assume the target bounding boxes parameters—specifically, the center coordinates $c_x$, $c_y$ and the dimensions $w$ (width) and $h$ (height)—as being drawn from their respective Gaussian distributions. Each of these parameters is characterized by an unknown mean $\mu$ and an unknown variance $\sigma^2$. For each bounding box parameter, we aim to estimate the Gaussian prior for the unknown mean and the Inverse-Gamma prior for the unknown variance,

$$(y_1, ..., y_N)_t \sim \mathcal{N}(\mu_t, \sigma_t^2), \text{ for } t \in \{c_x, c_y, w, h\}, \tag{11}$$

$$\mu_t \sim \mathcal{N}(\gamma_t, \sigma_t^2 v_t^{-1}), \quad \sigma_t^2 \sim \Gamma^{-1}(\alpha_t, \beta_t),$$

where $\Gamma(.)$ is the gamma function, $m = (\gamma, v, \alpha, \beta)$, and $\gamma \in \mathbb{R}, v > 0, \alpha > 1, \beta > 0$. For every bounding box parameter, our aim is to estimate a posterior distribution $q(\mu, \sigma^2) =$

$p(\mu, \sigma^2 | y_1, ..., y_N)$ which we can approximate as the Gaussian conjugate prior, *i.e.,* the Normal Inverse Gamma (NIG) Amini et al. (2020) as follows:

$$p(\underbrace{\mu, \sigma^2}_{\theta} \,|\, \underbrace{\gamma, \upsilon, \alpha, \beta}_{m}) = \frac{\beta^\alpha \sqrt{\upsilon}}{\Gamma(\alpha)\sqrt{2\pi\sigma^2}} \left(\frac{1}{\sigma^2}\right)^{\alpha+1} \exp\left\{-\frac{2\beta + \upsilon(\gamma-\mu)^2}{2\sigma^2}\right\}. \qquad (12)$$

The evidential regression loss, proposed by Amini et al. (2020), can be defined as:

$$\mathcal{L}_i^{\text{NLL}}(\Theta) = \tfrac{1}{2}\log\left(\tfrac{\pi}{\upsilon}\right) - \alpha\log(\Omega) + \left(\alpha + \tfrac{1}{2}\right)\log((y_i - \gamma)^2\upsilon + \Omega) + \log\left(\tfrac{\Gamma(\alpha_t)}{\Gamma(\alpha+\frac{1}{2})}\right), \qquad (13)$$

$$\mathcal{L}_i^{\text{R}}(\Theta) = |y_i - \gamma| \cdot (2\upsilon + \alpha), \qquad (14)$$

$$\mathcal{L}_i(\Theta) = \mathcal{L}_i^{\text{NLL}}(\Theta) + \lambda\,\mathcal{L}_i^{\text{R}}(\Theta), \qquad (15)$$

where $\Omega_t = 2\beta_t(1 + \upsilon_t)$. The bounding box then can be parameterized by the mean of the NIG distribution as $\hat{b} = \{\gamma_{cx}, \gamma_{cy}, \gamma_w, \gamma_h\}$. To account for different bounding box scales, we also incorporated a generalized IoU loss $\mathcal{L}_{iou}(., .)$ (Rezatofighi et al., 2019) that is scale-invariant. The overall evidential bounding box regression loss can be formulated as:

$$\mathcal{L}_{\text{bbox}} = \sum_{i=1}^{N} \left(\mathcal{L}_i(\Theta) + \lambda_{\text{IoU}}\mathcal{L}_{\text{IoU}}(b_i, \hat{b}_i)\right), \qquad (16)$$

where $b_i$ is the ground truth bounding box.

### 3.3 UNCERTAINTY AWARE OBJECT DETECTION

We extend DETR (Carion et al., 2020), a transformer-based object detection network, into an uncertainty-aware framework by integrating evidential deep learning into its architecture. However, our approach is easily extensible and can be applied to any subsequent DETR iterations (Zhu et al., 2020; Zhao et al., 2024). DETR models leverage the attention mechanisms in the encoder-decoder architecture of transformers for the task of detection, treating it as an unordered sequence, *i.e.* a set prediction problem, rather than relying on traditional methods that involve region proposal networks and non-maximum suppression.

Fig 1 illustrates our network architecture. The transformer decoder consists of multiple decoder layers where each layer outputs an estimation. Each decoder layer contributes to progressively refining the object query, allowing the model to better manifest the relationships between objects and their contexts in the image.

For localization, our aim is to learn the parameters of the higher-order, evidential distribution. We assume the estimates are drawn from the same Gaussian, and as we gather more evidence over multiple layers, for every layer, we iteratively refine the parameters of the Normal Inverse-Gamma distribution (NIG). For bounding box estimates given a NIG distribution, we can compute the prediction, aleatoric, and epistemic uncertainty as follows:

$$\underbrace{\mathbb{E}[\mu] = \gamma,}_{\text{prediction}} \qquad \underbrace{\mathbb{E}[\sigma^2] = \tfrac{\beta}{\alpha-1},}_{\text{aleatoric}} \qquad \underbrace{\text{Var}[\mu] = \tfrac{\beta}{\upsilon(\alpha-1)}}_{\text{epistemic}}. \qquad (17)$$

For classification, the aleatoric uncertainty can be defined as the variance of the beta distribution which can be calculated as follows:

$$\text{Var}(\theta_k) = \frac{\alpha_k \beta_k}{(\alpha_k + \beta_k)^2(\alpha_k + \beta_k + 1)}. \qquad (18)$$

To quantify epistemic uncertainty, we consider the ensemble of predictions from multiple decoder layers. Each layer provides an estimate of the class probabilities, which can be treated as samples from a distribution of possible models. We compute the mutual information between the predictions and the model parameters (Steuer et al., 2002).

Table 1: In-Domain object detection performance and Detection-Expected Calibration Error (D-ECE) on the KITTI detection benchmark.

| Model | Car ↑ | | | Pedestrian ↑ | | | Cyclist ↑ | | | D-ECE ↓ |
|---|---|---|---|---|---|---|---|---|---|---|
| Baseline | Easy | Medium | Hard | Easy | Medium | Hard | Easy | Medium | Hard | |
| CenterNet | 95.2 | 87.3 | 79.0 | 76.6 | 61.1 | 52.5 | 73.5 | 54.4 | 48.6 | 5.8 |
| DETR | 6.2 | 11.6 | 14.8 | 14.0 | 18.4 | 17.8 | 10.0 | 8.0 | 8.2 | 87.0 |
| D-DETR | 93.6 | 84.2 | 77.4 | 70.0 | 60.2 | 54.0 | 66.0 | 47.4 | 44.8 | 9.4 |
| RT-DETR | 95.9 | 92.6 | 88.3 | 84.9 | 77.5 | 71.7 | 82.5 | 65.3 | 62.5 | 8.3 |
| **Evidential Deep Learning** | | | | | | | | | | |
| EvCenterNet | 96.1 | 88.0 | 86.5 | 74.9 | 66.0 | 58.0 | 82.1 | 63.4 | 58.1 | 4.5 |
| **Instance-Aware (Ours)** | | | | | | | | | | |
| DETR (+EDL) | 94.3 | 84.0 | 77.3 | 78.0 | 69.5 | 63.0 | 66.5 | 50.3 | 48.4 | 7.9 |
| D-DETR (+EDL) | 95.8 | 85.1 | 79.2 | 76.5 | 65.3 | 58.5 | 68.9 | 52.0 | 47.2 | 4.6 |
| RT-DETR (+EDL) | **97.0** | **93.9** | **89.8** | **88.1** | **80.4** | **74.0** | **85.3** | **68.3** | **65.8** | **2.7** |

**Uncertainty-aware ranking** DETR establishes a one-to-one correspondence and addresses a global optimization problem utilizing self-attention to generate a distinct bounding box for each prediction. During training, DETR employs Hungarian matching (Kuhn, 1955) which guarantees that each predicted object has a unique match with a ground truth label. However, this results in training instability and slow convergence. DETR operates with a fixed set of predetermined queries, which necessitates careful selection of a subset of these queries to ensure effective performance.

Several solutions have been proposed to address these challenges. Deformable DETR (D-DETR) (Zhu et al., 2020) introduced a multi-scale deformable attention mechanism, which enables the network to dynamically sample locations based on the input features. RT-DETR (Zhao et al., 2024) further iterates the approach by incorporating query denoising techniques and implementing an uncertainty-minimal query selection scheme.

In this work, we present a probabilistic framework for Hungarian matching. In situations where multiple predictions may correspond to a single ground truth, evidential learning aids in resolving ambiguities by offering a probabilistic approach. The model evaluates the likelihood of each potential match, enabling it to make more informed decisions regarding the association of predictions with ground truths. By providing richer information about uncertainty, evidential learning enhances the training signals, leading to improved convergence and performance in the matching process. Consequently, the model becomes more adept at distinguishing between confident and uncertain predictions

## 4 EVALUATION

We incorporate our instance-aware uncertainty framework into DETR (Carion et al., 2020), Deformable-DETR (Zhu et al., 2020), and RT-DETR (Zhao et al., 2024). We use EvCenterNet (Nallapareddy et al., 2023) as our evidential deep learning baseline, along with Cal-DETR (Munir et al., 2024) and BPC (Munir et al., 2023) as our calibration baselines.

**Datasets** The networks are trained on KITTI (Geiger et al., 2012) and Cityscapes (Cordts et al., 2016). To demonstrate their generalization, out-of-domain evaluation is conducted on BDD100K (Yu et al., 2020), nuImages (Caesar et al., 2020), and Foggy Cityscapes (Sakaridis et al., 2018). The KITTI dataset consists of 3,712 images for training and 3,769 for evaluation and includes 3 classes: *car*, *pedestrian*, and *cyclist*. The Cityscapes dataset consists of 2,975 training images and 500 validation images, including 8 classes: *person*, *rider*, *car*, *truck*, *bus*, *train*, *motorbike*, and *bicycle*. BDD100K includes 10,000 images for the validation set and features the same classes as Cityscapes. Foggy Cityscapes is a foggy version of Cityscapes, which includes 500 validation images simulated with severe fog, used for evaluation in an out-of-domain scenario. The nuImages validation set is composed of 3,249 images, and the *car* and *pedestrian* classes are used for evaluation.

Table 2: Out-Domain object detection performance trained on KITTI and evaluated on BDD100k and nuImages demonstrating improved generalization on COCO $AP_{50}$ metric.

| Model | BDD100k ($AP_{50}$) | | nuImages ($AP_{50}$) | | Model | BDD100k ($AP_{50}$) | | nuImages ($AP_{50}$) | |
|---|---|---|---|---|---|---|---|---|---|
| Baseline | Car ↑ | Pedestrian ↑ | Car ↑ | Pedestrian ↑ | Evidential Deep Learning | Car ↑ | Pedestrian ↑ | Car ↑ | Pedestrian ↑ |
| CenterNet | 30.8 | 19.2 | 44.3 | 23.7 | EvCenterNet | 33.4 | 23.3 | 46.5 | 26.3 |
| DETR | 8.0 | 3.1 | 12.3 | 4.2 | DETR (+EDL) | 35.7 | 17.9 | 44.8 | 24.6 |
| D-DETR | 31.6 | 23.1 | 40.6 | 23.4 | D-DETR (+EDL) | 37.2 | 28.6 | 47.3 | 27.9 |
| RT-DETR | 41.6 | 30.9 | 48.7 | 30.5 | RT-DETR (+EDL) | **43.6** | **36.0** | **54.2** | **38.2** |

Table 3: Object detection and calibration performance against baseline calibration methods on Deformable-DETR (D-DETR) measured against D-ECE and COCO AP.

| Model | In-Domain (Cityscapes) | | | Out-Domain (Foggy Cityscapes) | | | Out-Domain (BDD100k) | | |
|---|---|---|---|---|---|---|---|---|---|
| | D-ECE ↓ | mAP ↑ | $mAP_{50}$ ↑ | D-ECE ↓ | mAP ↑ | $mAP_{50}$ ↑ | D-ECE ↓ | mAP ↑ | $mAP_{50}$ ↑ |
| D-DETR | 13.8 | 26.8 | 49.5 | 19.5 | 17.3 | 29.3 | 11.7 | 10.2 | 21.9 |
| BPC | 9.9 | 26.8 | 48.7 | 12.5 | 17.7 | 30.2 | 10.6 | 11 | 23.6 |
| Cal-DETR | **8.4** | 28.4 | 51.4 | 11.9 | 17.6 | 29.8 | 11.4 | 11.1 | 23.9 |
| D-DETR (+EDL) | 9.1 | **35.7** | **56.8** | **10.2** | **26.9** | **40.7** | **10.7** | **18.3** | **32.9** |

**Evaluation metrics** We utilize the KITTI evaluation benchmark to assess in-domain object detection performance on the KITTI dataset, and we use the COCO Average Precision (mAP) (Lin et al., 2014) for other datasets. Additionally, we employ detection expected calibration error (D-ECE) (Kuppers et al., 2020) to evaluate network precision error.

## 4.1 QUANTITATIVE RESULTS

Our uncertainty-aware network improves upon the baseline methods and effectively generalizes to more diverse and imbalanced datasets. In Table 1, the KITTI benchmark differentiates between 'easy' detections and more challenging detections that involve heavy occlusions and varying lighting conditions. Our instance-aware framework demonstrates a substantially lower D-ECE error, indicating that the network is both accurate and more confident in its predictions and can better navigate the challenges posed in real-world driving scenarios. DETR relies on the computationally expensive operation of self-attention in transformers and has necessitated various optimization techniques, such as multi-scale inputs, deformable attention and query denoising in subsequent iterations. Our efforts show that incorporating a self-calibrating probabilistic framework during training helps modulate self-attention within transformer networks and has the potential to be expanded to other attention-based networks.

Table 2 demonstrates that our instance-aware uncertainty framework can generalize effectively, even when fine-tuned on only a few thousand images. Compared to the previous evidential deep learning baseline, EvCenterNet, our method incorporates uncertainty within transformer networks, and our instance-aware weighting scheme shows a clear improvement for the underrepresented *Pedestrian* class, which had a heavily imbalanced training dataset with 10,608 instances of cars and only 1,654 instances of pedestrians.

In Table 3, we train on Cityscapes and evaluate on Foggy Cityscapes and BDD100K. All datasets share the same overall classes. Both BPC (Munir et al., 2023) and Cal-DETR (Munir et al., 2024) leverage train-time calibration losses and rely on a large amount of data to effectively improve detection performance. By comparison, our method significantly improves generalization and detection performance. We achieve a higher detection performance and the drop when moving from Cityscapes to Foggy Cityscapes is comparable to similar methods and our method generalizes much better when evaluated on BDD100K.

**Ablation and analysis** In Table 4, we conduct an ablation study on the novel components introduced in the paper. The original implementation proposed by Sensoy et al. (2018) cannot be easily extended to dense object detection, and the evidential framework fails against the overwhelming class-imbalanced datasets. Our IoU-aware weighing mechanism modulates the foreground-

Table 4: Ablation study on Deformable DETR on the various components introduced in the paper against D-ECE and COCO AP. The networks were trained for 12 epochs.

| Model | Cityscapes | | Cityscapes → BDD100k | | KITTI | | KITTI → BDD100k [Car Only] | |
|---|---|---|---|---|---|---|---|---|
| | D-ECE ↓ | AP ↑ | D-ECE ↓ | AP ↑ | D-ECE ↓ | AP ↑ | D-ECE ↓ | AP ↑ |
| **D-DETR** | 16.3 | 21.7 | 12.6 | 6.4 | 11.4 | 31.2 | 13.7 | 21.4 |
| **D-DETR + Standard EDL** | 78.3 | 2.4 | 83.6 | 0.8 | 82.6 | 11.4 | 23.7 | 4.2 |
| **D-DETR + IoU Aware Weights** | 13.6 | 27.4 | 14.2 | 10.6 | 9.6 | 38.3 | 10.4 | 23.4 |
| **D-DETR + Self-Calibration Regularization** | 9.6 | 30.5 | 11.7 | 14.3 | 4.9 | 43.2 | 5.3 | 26.9 |
| **D-DETR + Hungarian Matching** | 9.5 | 30.9 | 11.7 | 14.4 | 4.9 | 43.1 | 5.2 | 27.1 |

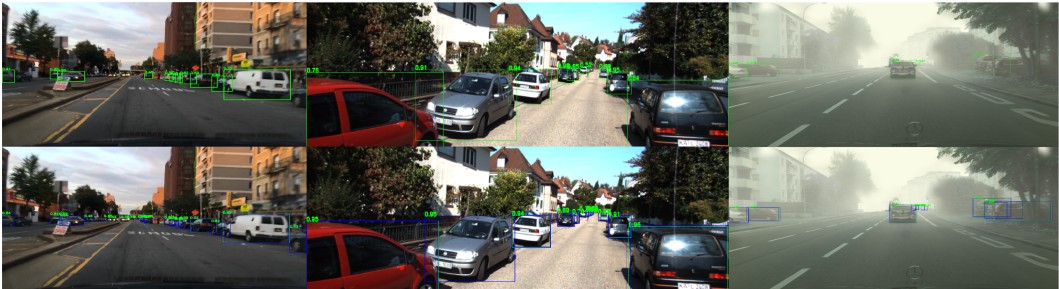

Figure 2: Qualitative results from Deformable-DETR on BDD100K, KITTI, and Foggy Cityscapes, in that order. The top figures represent the baseline D-DETR, while the bottom figures incorporate our uncertainty framework. The green lines indicate the mean, while the blue lines represent uncertainty within the width and height at different confidence thresholds. The centers are kept at their mean values without applying uncertainty estimates.

background imbalance and evidential deep learning is very effective in improving generalization of detection performance. However, the adversarial nature of forcing the model to remain uncertain diminishes the true potential of the evidential framework. By enabling the model to be self-aware of its uncertainty, it is able to implicitly calibrate its performance and improve confidence in its predictions. Finally, we integrate the estimated uncertainties as an application within the network itself and utilize them to address the global optimization problem for DETR models, which are designed to learn one-to-one correspondences for detections.

## 4.2 QUALITATIVE RESULTS

In Figure 2, the qualitative results demonstrate that, compared to the baseline D-DETR, our uncertainty-aware model exhibits significantly greater confidence in its predictions, particularly for occluded and partially out-of-view vehicles. Furthermore, we can predict distributions of bounding boxes and learn their variance to effectively model uncertainty for downstream applications. Nevertheless, due to our IoU-aware weighting scheme, confidence and IoU are correlated; generally, only bounding boxes with high IoU and low variance will exhibit high confidence.

## 5 CONCLUSIONS

The results are quite impressive and highly applicable to many real-world scenarios where pretrained networks are fine-tuned on datasets with limited samples, particularly in safety-critical applications. The calibration performance is comparable to that of other state-of-the-art methods, while our approach also improves detection performance.

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

# A    APPENDIX

## A.1    EXPERIMENTAL SETUP

**Datasets and metric**  The network was trained on KITTI and Cityscapes and evaluated on KITTI, Cityscapes, Foggy Cityscapes, BDD100K, nuImages. For evaluation, Cityscape and Foggy Cityscapes were resized to [1024, 512] from [2048, 1024], and BDD100K and nuImages to a resolution of [896, 512] from [1280, 720] and [1600, 900] respectively. We used the official KITTI benchmark, pycocotools, and netcal for evaluating KITTI metrics, COCO metrics and D-ECE. D-ECE was evaluated using a bin size of 10, IoU threshold of 0.5, and score threshold of 0.3.

The network was trained on a single GPU with 24GB VRAM.

All baseline models followed their respective training schemes. All Evidential DETR models followed a common training scheme unless stated otherwise so they could also be directly compared against each other.

**Implementation details**  We used Resnet-50 backbone for all the different DETR models and used the default COCO weights the models were originally trained on. The default architecture parameters were used and the following is a list of shared hyperparameters.

Table 5: E-DETR shared parameters.

| Item | Value |
|---|---|
| epochs | 24 |
| optimizer | AdamW |
| base learning rate | 1e-4 |
| backbone learning rate | 1e-5 |
| freezing BN | True |
| weight decay | 1e-3 |
| clip gradient norm | 0.1 |
| $\lambda_{ann}$ start | 0.0 |
| $\lambda_{ann}$ end | 0.1 |
| $\lambda_{ann}$ linear decay steps | 7420 |
| lr decay rate | 0.1 |
| lr step size | 7420 |
| cudnn benchmark | True |

## A.2 Extra evaluations

Table 6: COCO vs Corrupted COCO on E-DETR (D-DETR) trained for 12 epochs measured against D-ECE and COCO AP.

| COCO | | Corrupted COCO | |
|---|---|---|---|
| D-ECE ↓ | mAP ↑ | D-ECE ↓ | mAP ↑ |
| 10.3 | 38.3 | 10.7 | 14.7 |

Table 7: Cityscapes vs Corrupted Cityscapes on E-DETR (D-DETR) trained for 12 epochs measured against D-ECE and COCO AP.

| Cityscapes | | Corrupted Cityscapes | |
|---|---|---|---|
| D-ECE ↓ | mAP ↑ | D-ECE ↓ | mAP ↑ |
| 9.6 | 30.5 | 9.9 | 12.6 |

Table 8: Sim10K vs BDD100K (Car) on E-DETR (D-DETR) trained for 12 epochs measured against D-ECE and COCO AP.

| Model | Sim10K | | BDD100K | |
|---|---|---|---|---|
| | D-ECE ↓ | AP ↑ | D-ECE ↓ | AP ↑ |
| BCH | 6.1 | 65.4 | 6.3 | 23.4 |
| Cal-DETR | 6.2 | 65.9 | 6.3 | 23.8 |
| E-DETR | 7.4 | 61.3 | 7.2 | 20.6 |

