# OpenReview forum: "E-DETR: Evidential Deep Learning for End-to-End Uncertainty Estimation in Object Detection"
_ICLR.cc/2025/Conference — ICLR 2025 Conference Withdrawn Submission_

### Official Review · Reviewer_krKg · 2024-10-15

**Soundness:** 3
**Presentation:** 1
**Contribution:** 2
**Rating:** 3
**Confidence:** 4

**Summary:**

The proposed work presents an uncertainty estimation method for object detection. Deep evedential learning is used for learning a posterior distribution. The approach is sampling free. An instance-aware uncertainty framework by extending evidential deep learning with an IoU-aware loss, jointly modelling both classification and localization uncertainties is proposed.

**Strengths:**

1) The paper's narration is easy to follow.

2) The results that were provided are convincing.

3) Uncertainty estimation for object detection is challenging and important. However, there has been a number works published previously (only few of them are cited).

4) The proposed combination of mostly existing pieces is well-designed.

**Weaknesses:**

1) The organization of the manuscript offers room for improvement. The section on the numerical experiments is short, the method section lengthy, the number of citations does not reflect the existing body of related work.

2) The amount of numerical experiments is thin. How does the method react to semantic out-of-distribution examples? How does it react to adversarial attacks? I find that a consideration of ECE only is not enough. How does the proposed method compare to confidence calibration methods? How does the calibration behave in the study provided in Tab. 2?

3) It is good that the primary object detection performance improves over other methods in Tab. 1 and also the results provided are convincing. However, primary accuracy + ECE are not enough for evaluating an uncertainty estimate in my opinion. How well does the uncertainty correlate with the presence of errors? How does it behave on semantic out-of-distribution objects or adversarial attacks?

Concerns of higher degree of detail:

4) No citations in the paragraphs 1, 2, 4 although several claims are made, such as overconfidence of object detectors etc. In general, the density of citations could be improved. Existing related works like [1,2] have not been considered.

5) The example provided for cross-entropy + softmax suffering from overconfidence, namely rotations applied to the input causing overconfidence, is a general problem caused by domain shift. Is this really mitigated by the proposed approach? With which data is the "I don't know" regularization term trained? Just using the background? Could the authors elaborate on that?

6) In eq. (8) the IoU is introduced in terms of q_ij. An IoU of zero does not provide any learning signal. Could the authors clarify on that?

7) In my opinion, it could be pointed out a bit more strongly which parts of the method are novel.

8) The claim of proposing a highly generalizable uncertainty estimation method seems too strong as it is at the same time proposed only for DETR models

9) The claim that the proposed method "align[s] with the true likelihood" seems a bit strong to me. Can the authors clarify what they mean?

10) Table 1 -- besides ECE, the type of evaluation metric (AP?) could be mentioned in the table.

In opinion, in the current state, the manuscript is not ready for publication at a conference like ICLR.

[1] Miller, Dimity, et al. "Dropout sampling for robust object detection in open-set conditions." 2018 IEEE International Conference on Robotics and Automation (ICRA). IEEE, 2018.

[2] Gasperini,  Stefano et al. "CertainNet: Sampling-free Uncertainty Estimation for Object Detection" IEEE ROBOTICS AND AUTOMATION LETTERS. 2021.

[3] Riedlinger, Tobias, et al. "Gradient-based quantification of epistemic uncertainty for deep object detectors." Proceedings of the IEEE/CVF Winter Conference on Applications of Computer Vision. 2023.

**Questions:**

1) Could you clarify what is the benefit of utilizing evidential deep learning in comparison to other approaches for uncertainty estimation in object detection compared to other approaches like e.g. [2,3] that are sampling free as well? In particular [3] is a post-hoc method which is more widely applicable, can also be traded in for performance and provides calibrated confidences.

2) Nallapareddy et al. (2023) receives a significant portion of the related work section. How is this approach related to the proposed work?

3) Could the authors comment on the questions in weaknesses 2,3,5,6,9?

4) Could the authors explain how they would like to address the remaining weaknesses 1,4,7,8,10?

**Details Of Ethics Concerns:**

Does not apply.

---

> ### Author Response · Authors · 2024-11-29
>
> **Q: Could you clarify what is the benefit of utilizing evidential deep learning in comparison to other approaches for uncertainty estimation in object detection compared to other approaches like e.g. [2,3] that are sampling free as well? In particular [3] is a post-hoc method which is more widely applicable, can also be traded in for performance and provides calibrated confidences.**
>
> A: CertainNet [2] employs a computationally expensive RBF kernel for DUQ and has already been outperformed by the comparative baseline that we surpass in our work, namely EVCenterNet [Nallapareddy et al. (2023)].
>
> The cited WACV 2023 work [3] focuses on epistemic uncertainty only.
> In contrast, our approach utilizes evidential deep learning to directly estimate the posterior distribution, and our method allows us to quantify both epistemic and aleatoric uncertainty.  Furthermore, uncertainty quantification in gradient-based methods relies on the gradients and is typically sensitive to, and limited by, the training sample set size and class imbalance. Our method is able to generalize effectively and calibrate confidence even when fine-tuned on small datasets. We demonstrate good results even when working with datasets under 5k images.
>
> Our method only requires fine-tuning, leverages its uncertainty for self-calibration, and does it all without any significant reduction in performance.
>
> *Valdenegro-Toro, Matias. "Exploring the limits of epistemic uncertainty quantification in low-shot settings." arXiv preprint arXiv:2111.09808 (2021).*
>
> *Choi, Ching Lam, and Farzan Farnia. "On the Generalization of Gradient-based Neural Network Interpretations."*
>
> **Q: Nallapareddy et al. (2023) receives a significant portion of the related work section. How is this approach related to the proposed work?**
>
> A: Nallapareddy et al. (2023) previously successfully applied evidential deep learning to CenterNet, a key-point based object detection which regresses object centers as Gaussian heatmaps.
> It is important to clearly detail this method, as it stands out as a recent (2023), relevant and  competitive baseline against which to evaluate our work. Note that their method requires additional regularization and is implicitly conditioned on the resolution of the heatmap and requires post-processing. In contrast, our method is completely end-to-end, self-calibrating and without need for post-processing.
>
> **Q: The organization of the manuscript offers room for improvement... In my opinion, it could be pointed out a bit more strongly which parts of the method are novel..."**
>
> A: Thank you for this feedback. We have revised the methodology and improved the linkage between contributions in the introduction, the methodology, and the ablation study.
>
> **Q: No citations in the paragraphs 1, 2, 4 although several claims are made... Existing related works like [1,2] have not been considered...**
>
>  A: Thank you for your feedback regarding missing citations. We have added more citations to back our claims. Also, we have tried to limit the scope to only evidential deep learning and DETR based works.
>
> **Q: The claim of proposing a highly generalizable uncertainty estimation method seems too strong as it is at the same time proposed only for DETR models.**
>
> A: Our method is only suitable for DETR models, and by generalization we were referring to the data and generalizing to large diverse datasets while fine-tuning on only small datasets with under 5k images.
>
> **Q: Table 1 -- besides ECE, the type of evaluation metric (AP?) could be mentioned in the table.**
>
> A: We have revised the paper to clarify the use of D-ECE and COCO AP where applicable.

---

> ### Author Response · Authors · 2024-11-29
>
> **Q: The amount of numerical experiments is thin. How does the method react to semantic out-of-distribution examples?**
>
> A: We have included additional evaluations, training on Cityscapes and evaluating on out-of-distribution of Foggy Cityscapes and BDD100k and also show comparisons against Cal-DETR, a calibration baseline that also leverages uncertainty for calibration.
>
> **Q: How does it react to adversarial attacks?**
>
> A: Our primary motivation was to leverage uncertainty to improve generalization on small datasets; however, we understand the importance of addressing adversarial attacks as well. We have included additional evaluations that focus on image corruption. While we observed a significant drop in detection performance, the calibration results remain strong. We believe that effectively addressing adversarial attacks would require modifications to the training scheme to enhance robustness against adversarial augmentations.
>
> **Q: The example provided for cross-entropy + softmax suffering from overconfidence, namely rotations applied to the input causing overconfidence, is a general problem caused by domain shift. Is this really mitigated by the proposed approach?**
>
> A: Rotations are typically not well suited as augmentations for axis-aligned bounding boxes in traditional object detection. This would change the shape and scale of objects beyond the priors learned in training.
>
> **Q: How well does the uncertainty correlate with the presence of errors?...  With which data is the "I don't know" regularization term trained? Just using the background?... The claim that the proposed method "align[s] with the true likelihood" seems a bit strong to me. Can the authors clarify what they mean?**
>
> A: We have revised the methodology to better clarify the learning of the Beta distribution. There is no explicit background class and for each class, all other classes are also treated as part of the background.
>
> DETR models follow a one-to-one mapping and for each ground truth object only one prediction can be mapped to it. For all matched predictions, we model individual Beta distributions for all classes which converts the problem into a binary classification of "object" and "not object". Our uncertainty regularization method also includes a self-calibrating hyperparameter based on calibration error. In evidential deep learning, the Beta distribution outputs the evidence for the class, when the object belongs to the class the classification error is reduced when new evidence is added and decreased when evidence is removed, aligning it with the true likelihood.
>
> The network outputs $\alpha$ and $\beta$ for all classes which effectively represent the pseudo count for that class. E.g The network inputs an image of a cat and directly outputs a distribution that given the feature representation 10 times it thought it was a cat and 5 times it was not a cat. This is normalized to output the final probability. The confidence is then further used for uncertainty regularization and model calibration.
>
> **Q: In eq. (8) the IoU is introduced in terms of q_ij. An IoU of zero does not provide any learning signal. Could the authors clarify on that?**
>
> A: During training, DETR models employ Hungarian matching to solve a global optimization problem between the predictions and ground truth for a one-to-one mapping. Our contribution also introduces a probabilistic framework to this optimization process, incorporating uncertainty in the matching process.
>
> The global optimization process typically ensures there is no zero IoU. While the IoU-aware loss helps integrate localization accuracy towards overall confidence, it is still a multi-task framework and the IoU will be maximized during training as a separate task.

---

> > ### Comment · Reviewer_krKg · 2024-11-29
> >
> > I'd like to thank the authors for the detailed response. Some of my questions have been answered. However, it was mostly not indicated that the manuscript will be revised. Although I acknowledge the effort of the authors, the new experiments to do not help me much for my understanding. They lack baseline comparisons. Also I don't agree that narrowing down the literature review to evidential deep learning is reasonable. When going to object detection, the application is of importance as well. In my opinion it matters a lot in the end, which approach is most useful in practice. Still I am missing insights on the usefulness of the proposed uncertainties. Do they help to find false positives, and OOD objects. Thus, I stick to my initial rating.

---

### Official Review · Reviewer_PNF3 · 2024-11-02

**Soundness:** 2
**Presentation:** 2
**Contribution:** 2
**Rating:** 5
**Confidence:** 3

**Summary:**

This paper extends the DETR framework by replacing point estimate regressions with direct learning of the posterior distribution, leveraging deep evidential learning. The method presents an instance-aware uncertainty framework that jointly models classification and localization uncertainties through an IoU-aware loss. It also enables the model to use its uncertainty for self-calibration, aligning predicted probabilities with true outcomes.

**Strengths:**

1.The motivation is clear and convincing, effectively identifying the current challenges and deficiencies in existing methods, and proposing targeted modular solutions. The results achieved in experiments are commendable.

2.The method shows a certain degree of novelty by applying evidential deep learning to enhance DETR and innovatively presenting an instance-aware uncertainty framework.

3.The paper is well-supported theoretically, using extensive citations and formulas to demonstrate the robustness of the approach.

**Weaknesses:**

1.The illustrations in the paper are subpar. There is only one "rudimentary" network diagram with very limited information. Although the approach is theoretically inclined, I believe that incorporating high-quality images—such as diagrams, module schematics, and charts analyzing experimental data—is essential for a good paper, as they effectively aid reader comprehension.

2.The paper contains numerous formulas that lack punctuation at the end. A thorough review is recommended.

3.The formatting of tables in the paper appears unconventional, with a significant amount of whitespace.

4.The experimental results are insufficient and somewhat confusing. First, I suggest that the authors conduct additional experiments to further verify their methodology, such as hyperparameter discussions and experiments addressing the identified challenges. Additionally, the accuracy reported for DETR in Tables 1 and 2 seems unusual; I believe these accuracies are unreasonable and recommend that the authors carefully review the implementation of DETR. The specific implementation details for Table 3 are also not described.

5.While extensive use of formulas to argue the method can be persuasive, including all of them in the main text does not facilitate a quick and deep understanding for the reader. I recommend adjusting the description of the method section to emphasize the improvements specific to this work, while placing formulas that in existing other works in the supplementary materials.

**Questions:**

see Weaknesses

---

> ### Author Response · Authors · 2024-11-29
>
> **Q: The experimental results are insufficient and somewhat confusing. First, I suggest that the authors conduct additional experiments to further verify their methodology...**
>
> A: We have included additional results in the revision and expanded the ablation study.
>
> **Q:  ...such as hyperparameter discussions and experiments addressing the identified challenges.**
>
> A: We have included the hyperparameters used in the appendix. The sample space for these hyperparameters is extensive, and a study is still underway to investigate their impact.
>
> **Q: Additionally, the accuracy reported for DETR in Tables 1 and 2 seems unusual; I believe these accuracies are unreasonable and recommend that the authors carefully review the implementation of DETR**
>
> A: We have verified the implementation on our end since ours is a very minor change to the original code. We will release our code at a later date for public scrutiny.
>
> **Q: The specific implementation details for Table 3 are also not described.**
>
> A: This has been fixed in the revision.
>
> **Q: While extensive use of formulas to argue the method can be persuasive, including all of them in the main text does not facilitate a quick and deep understanding for the reader. I recommend adjusting the description of the method section to emphasize the improvements specific to this work, while placing formulas that in existing other works in the supplementary materials.**
>
> A: We have removed the equations that weren't being directly used in the paper and added citations for them. We have slightly revised the methodology section, and the contributions outlined in the introduction, methodology, and ablation studies are now more cohesively linked.
>
> **Illustrations:** We have improved the network architecture figure to better illustrate how our model outputs distributions. Additionally, we have included qualitative comparisons to improve the clarity and impact of our results.
>
>
> **Formatting:** Thank you for the feedback regarding formatting. We have reviewed the punctuation in the equations and reduced the whitespace in the tables. The tables are hopefully formatted to align with the ICLR guidelines.
>
> “Place one line space before the table title, one line space after the table
> title, and one line space after the table.

---

> > ### Comment · Reviewer_PNF3 · 2024-11-29
> >
> > Thank you very much for the author's response. However, I regret to say that I did not glean any substantive content regarding the improvement of the paper's quality from the author's reply, and I will maintain my rating.

---

### Official Review · Reviewer_o9um · 2024-11-02

**Soundness:** 3
**Presentation:** 2
**Contribution:** 2
**Rating:** 5
**Confidence:** 3

**Summary:**

The paper tackles the problem of jointly estimating classification and localization uncertainty estimates in an end-to-end manner for transformer-based detection methods (DETR). The core idea is to extend the deep evidential learning by introducing an IoU-aware loss, thereby jointly modeling the classification and localization uncertainties. The estimated uncertainties are used to self-calibrate the model which aligns the predicted probability with the true likelihood of outcomes. Experiments have been conducted with different variants of DETR. The results have been shown by training on KITTI dataset and evaluating on BDD-100K and nuImages datasets. Reported results claim to surpass the baseline methods and competing methods by notable margins.

**Strengths:**

**1)** The problem of estimating the classification and localization uncertainty estimation for object detection task carries importance due to its several applications in safety- critical applications.

**2)** The paper proposes an IoU-aware loss to extend deep evidential learning to jointly model the classification and localization uncertainties.

**3)** The uncertainty estimates allows self-calibration of the detection model, which is important to develop trust in its predictions and improve the overall reliability of the model.

**4)** Experiments have been shown with different variants of DETR and the results claim to significantly outperform the baselines and competitive methods.

**Weaknesses:**

**1)** The scale of experiments is rather small, especially across the diversity and number of datasets, and so it is difficult to fully accept the effectiveness of the method.

**2)** The ablation study is not deep and also conducted on only one dataset and with one baseline detection method.

**3)** Why the expected calibration error (ECE) is used to report the calibration performance of detection methods? It is difficult to understand as it is primarily for reporting classification performance and there are proper metrics such as D-ECE [A], LaECE [B] for revealing the calibration performance of detectors.

**4)** The method section is a bit hard to read because the core contributions of the paper are somewhat hidden.

**5)** The paper claims at several places (e.g., L22, L88, L151) of handling the class imbalance problem, however, there is no dedicated study in the paper which validates this clearly.

**6)** The paper is missing insights on why and how it is able to substantially boost the performance of transformer-based detection methods, especially the DETR method.

[A] Küppers, F., Haselhoff, A., Kronenberger, J. and Schneider, J., 2022. Confidence calibration for object detection and segmentation. In Deep Neural Networks and Data for Automated Driving: Robustness, Uncertainty Quantification, and Insights Towards Safety (pp. 225-250). Cham: Springer International Publishing.

[B] Kuzucu, S., Oksuz, K., Sadeghi, J. and Dokania, P.K., 2024. On Calibration of Object Detectors: Pitfalls, Evaluation and Baselines. arXiv preprint arXiv:2405.20459.

**Questions:**

**1)** I would like to see the results (detection performance and calibration performance) on more in-domain and out-domain scenarios, such as Cityscapes (CS) and CS to CS-foggy, Sim10K and Sim10K to BDD100K (same classes as Sim10K. In BDD100K), and COCO.

**2)** Detailed ablation studies to establish the importance of proposed components across 2 more scenarios of in-domain and out-domain.

**3)** It is very important to investigate the performance (both detection and calibration) in different class-imbalance scenarios. Further, I expect to see the performance under dense detection scenarios (other than BDD100K and nuImages) as claimed by the method.

**4)** L484: The paper mentions that, it is able to significantly improve the DETR performance comparable to newer versions of DETR. My question is how is this result relevant, when the achieved gain when later variants of DETR such as Deformable-DETR itself Is well over three years old.

**5)** How do you set $\lambda_{ann}$ in eq(10).

**6)** It would be good to show that how capable is proposed method in overcoming overconfidence and underconfidence in predictions.

**7)** The first para. of introduction section is missing references, for example L37 and L42.

**8)** Figure 1 Is not referenced anywhere in the text of the paper.

**9)** The paper needs to provide some insights in the form of analyses to better establish the grounding of claimed notable performance gains reported, especially in the case of DETR baseline

---

> ### Author Response · Authors · 2024-11-29
>
> **Q: I would like to see the results (detection performance and calibration performance) on more in-domain and out-domain scenarios, such as Cityscapes (CS) and CS to CS-foggy, Sim10K and Sim10K to BDD100K (same classes as Sim10K. In BDD100K), and COCO.**
>
> A: We have added additional results for in-domain on Cityscapes and out-domain to Foggy Cityscapes and BDD100k, Sim10K to BDD100K as well as COCO to Corrupted COCO. Furthermore, we have included more baselines for calibration on DETR models, Cal-DETR and BPC.
>
> **Q: Detailed ablation studies to establish the importance of proposed components across 2 more scenarios of in-domain and out-domain.**
>
> A: We have expanded the ablation studies to cover all our contributions across two scenarios. Cityscapes and Cityscapes to BDD100K, which share the same classes. And KITTI and KITTI
>  to BDD100K, focusing on the Car category.
>
> **Q: It is very important to investigate the performance (both detection and calibration) in different class-imbalance scenarios. Further, I expect to see the performance under dense detection scenarios (other than BDD100K and nuImages) as claimed by the method.**
>
> A: The datasets we have chosen to work with are heavily class imbalanced. The following are the frequencies on their validation sets.
>
>  **Category**    | **Cityscapes** | **BDD100K** | **KITTI**  |
> |-------------|------------|---------|--------|
> | Car         | 20,694     | 102,837 | 14,385 |
> | Bus         |    282     |  1,660  |  - |
> | Person      | 13,402     |  13,425 |   2,280|
> | Motorcycle   |    541     |    460  |   -    |
> | Rider       |  1,296     |    658  |   893    |
> | Bicycle     |  2,629     |  1,039  |   -    |
> | Truck       |    376     |  4,243  |   -    |
> | Train       |    123     |    15   |   -    |
>
> The standard implementation for evidential deep learning, designed for image classification, cannot be adopted for object detection easily. Our main contribution in this paper is we have introduced an instance-aware uncertainty estimation scheme for class balancing, enabling us to leverage evidential deep learning in DETR models. In the revision, we have also included additional evaluations and improved the ablation study.
>
> **Q: L484: The paper mentions that, it is able to significantly improve the DETR performance comparable to newer versions of DETR. My question is how is this result relevant, when the achieved gain when later variants of DETR such as Deformable-DETR itself Is well over three years old.**
>
> We recognize that different iterations of DETR have explored various approaches, such as deformable attention, multi-scale inputs, and denoising anchors. Our probabilistic uncertainty framework represents a novel contribution that significantly improves the performance of the original DETR model and its newer iterations.
>
> **Q: How do you set $\lambda_{ann}$ in eq(10).**
>
> A: We have included the hyperparameters used in the revised appendix. $\lambda_{ann}$ is updated linearly every iteration for 20 epochs. All E-DETR models were trained for 24 epochs.
>
> **Q: It would be good to show that how capable is proposed method in overcoming overconfidence and underconfidence in predictions.**
>
> A: The methodology section has been slightly revised to better explain how the model uses its uncertainty to deal with overconfidence and underconfidence. Model calibration has also been added to the ablation study.
>
> The network outputs $\alpha$ and $\beta$ for all classes which effectively represent the pseudo count for that class. E.g The network inputs an image of a cat and directly outputs a distribution that given the feature representation 10 times it thought it was a cat and 5 times it was not a cat. This is normalized to output the final probability. The confidence is then further used for uncertainty regularization and model calibration.
>
> **Q: The first para. of introduction section is missing references, for example L37 and L42.**
> **Q: Figure 1 Is not referenced anywhere in the text of the paper.**
>
> A: Thank you for pointing out the missing references. We have added citations to back our claims. Figure 1 has been improved considerably and is referred to appropriately in the paper now.
>
> **Q: The paper needs to provide some insights in the form of analyses to better establish the grounding of claimed notable performance gains reported, especially in the case of DETR baseline**
>
> A: The contributions, methodology, and ablation study should be all linked now.

---

> > ### Comment · Reviewer_o9um · 2024-11-29
> > **Response to author's rebuttal**
> >
> > I would like to thank authors for submitting a rebuttal to my comments and questions. After reading the rebuttal, I still believe that there are some important gaps in this study as listed below:
> >
> > - The submission and the rebuttal is missing analyses on establishing the grounding of the approach? The ablation study only provides a holistic understanding (i.e. numbers) of the impact of each claimed contribution without any much insights.
> >
> > - An important claim of the paper is improving model calibration, however, as per new results provided, the method struggles in reducing D-ECE in many cases compared to other baselines in datasets such as, Sim10K and BDD100K. Is there any specific reason for this? Also, why the results on these important datasets were not included at the time of paper submission?
> >
> > - Beyond D-ECE, why the other evaluation metrics such as LaECE are not used to report calibration performance?
> >
> > -  It is still not clear that how the capable is proposed method in reducing overconfidence/underconfidence. I was expecting some kind of plots such as, reliability diagrams and/or histograms showing confidence of incorrect predictions.

---

### Official Review · Reviewer_rcw5 · 2024-11-02

**Soundness:** 2
**Presentation:** 3
**Contribution:** 2
**Rating:** 5
**Confidence:** 4

**Summary:**

The work presents an approach to incorporate an evidential deep learning mechanism into the DETR architecture, replacing the regression of point estimates with the direct learning of posterior distributions. By leveraging an intersection-over-union (IoU) aware loss within an evidential deep learning framework, the method aims to provide instance-aware uncertainty estimates. This is particularly important for model calibration in safety-critical applications. The approach is mainly plug-and-play and can be extended to any DETR-based architecture. Results are demonstrated on the KITTI and BDD100k datasets.

**Strengths:**

- The proposed method introduces a straightforward modification that can be easily integrated into existing DETR-based models without adding significant complexity. This plug-and-play nature promotes adoption across various models.
- The paper tackles the important issue of uncertainty estimation in object detection, which is crucial for calibrating detectors in safety-critical applications.
- The structure of the paper is well organized.

**Weaknesses:**

- The experiments are limited to KITTI and BDD100k datasets. To effectively validate the usefulness of the method, it should be tested on relatively recent DETR-based architectures like DINO [1] and on larger, more diverse datasets such as MS-COCO. Exploring settings like DINO where models are pre-trained on Objects365 and fine-tuned on COCO would provide insights into the method's scalability.
- The paper does not include comparisons with prominent calibration methods that have shown improvements in DETR-based architectures [2,3]. Incorporating these comparisons highlights the advantages or limitations of the proposed approach.
- It is unclear how the Expected Calibration Error (ECE) is calculated. If Detection-ECE [4] is used, specifying the factors considered during computation (e.g., confidence scores, IoU thresholds, etc) is necessary.
- Additionally, recent metrics proposed in the literature, such as in [5,6], should be included to provide a comprehensive evaluation of calibration performance.
- Though the paper is well organized, some parts of the methodology are difficult to grasp.

[1] "DINO: DETR with Improved DeNoising Anchor Boxes for End-to-End Object Detection." ICLR (2023)

[2]  "Cal-DETR: calibrated detection transformer." Advances in neural information processing systems (2024)

[3] "Bridging precision and confidence: A train-time loss for calibrating object detection." Conference on Computer Vision and Pattern Recognition. (2023)

[4] "Multivariate confidence calibration for object detection." Conference on computer vision and pattern recognition workshops. (2020)

[5] "Towards building self-aware object detectors via reliable uncertainty quantification and calibration." Conference on Computer Vision and Pattern Recognition. (2023)

[6] "On Calibration of Object Detectors: Pitfalls, Evaluation and Baselines." ECCV (2024)

**Questions:**

- Experiments are currently limited to the KITTI and BDD100k datasets. Have authors considered evaluating method on larger and more diverse datasets like MS-COCO, and on more recent DETR-based architectures such as DINO? Extending the experimental validation could demonstrate the scalability and generalization capabilities of the approach.
- The paper does not include comparisons with prominent calibration methods that have shown improvements in DETR-based architectures. How does the proposed method perform relative to these existing approaches?

---

> ### Author Response · Authors · 2024-11-29
>
> **Q: The paper does not include comparisons with prominent calibration methods that have shown improvements in DETR-based architectures**
>
> A: Thank you for suggesting suitable calibration comparison methods for DETR. We have added comparisons against them in our revision (Table 3) and outperformed both methods on out-of-domain data, often by a substantial margin.
>
> **Q: To effectively validate the usefulness of the method, it should be tested on relatively recent DETR-based architectures like DINO [1]...**
>
> A: We have included RT-DETR in our evaluation. To the best of our knowledge, this is the latest iteration of DETR and incorporates the query denoising introduced in DINO. However, RT-DETR is faster to train and outperforms all existing DETR models and so we believe this is the strongest baseline against which to compare.
>
> **Q: …and on larger, more diverse datasets such as MS-COCO**
>
> A: We have added additional results for in-domain on Cityscapes and out-domain to Foggy Cityscapes and BDD100k as well as for COCO and Corrupted COCO. However, our motivation is to improve generalization when training on datasets with limited sample sizes. As the sample size decreases quantifying uncertainty becomes more challenging, however, this is also where evidential deep learning tends to shine, though, it's not suitable for object detection out of the box thus necessitating the paper. We have chosen to focus on datasets with less than 10,000 images with dense object detections and have primarily included self-driving datasets.
>
> **Q: It is unclear how the Expected Calibration Error (ECE) is calculated. If Detection-ECE [4] is used, specifying the factors considered during computation (e.g., confidence scores, IoU thresholds, etc) is necessary.**
>
> A: The reviewer is correct that we used Detection-ECE as our calibration error metric. In the revised paper we now state this clearly (and that we use COCO metrics for object detection). The hyperparameters for the metric have also been included in the appendix.
>
> **Q: recent metrics proposed in the literature, such as in [5,6], should be included to provide a comprehensive evaluation of calibration performance.**
>
> A: Since we do not have access to the weights for EVCenterNet (Nallapareddy et al., IROS 2023) and Cal-DETR (Munir et al., NeurIPS 2024), we are unable to conduct our own evaluations using the new metrics. Therefore, we have adopted the same metrics used by our baseline models as published in their respective papers.
>
> **Q: Though the paper is well organized, some parts of the methodology are difficult to grasp.**
>
> A: The methodology section has also been improved for clarification from line 230. The contributions in the introduction, the methodology, and the ablation studies should be linked now. The network architecture has been improved.

---

> ### Comment · Reviewer_rcw5 · 2024-11-30
>
> I thank the authors for their rebuttal. After reviewing their responses, I feel some areas still require further clarity and improvement. Specifically, the computation of D-ECE remains unclear, which significantly impacts the interpretability of the calibration results, given D-ECE's sensitivity to threshold levels. Reliability diagrams are missing, and including these diagrams to illustrate underconfidence and overconfidence would enhance the understanding of the calibration outcomes. Additionally, the consistent underperformance of E-DETR, as shown in the provided results, compared to BPC and Cal-DETR, raises concerns about its practical utility. I will maintain my original rating.

---

### Author Response · Authors · 2024-11-29
**Response to common concerns of all reviewers**

We thank all four reviewers for the detailed and thoughtful comments and questions. Three reviewers are on the borderline while one is more negative. However, all reviewers appreciate the importance of modelling uncertainty in object detection and appreciate the key values of our proposed method, e.g. “a straightforward modification that can be easily integrated into existing DETR-based models without adding significant complexity”, “allows self-calibration of the detection model, which is important to develop trust in its predictions and improve the overall reliability of the model” and “results that were provided are convincing”. There was a common concern that the experimental evaluation was not sufficient to support our claims. We have addressed this by including many new results and also summarize some of them here.

Below, we show generalization to new domains on E-DETR based on Deformable-DETR trained for 12 epochs: 1. Testing first on COCO and then Corrupted COCO, 2. Testing first on Cityscapes and then on Corrupted Cityscapes, 3. Testing first on SIM10K and then on BDD100k. For introducing corruptions we used the imagecorruptions library (Hendrycks, D., & Dietterich, T. (2019) ICLR).

|            | COCO      |         | Corrupted COCO |         |
| ---------- | --------- | ------- | -------------- | ------- |
| **Model**  | **D-ECE** | **mAP** | **D-ECE**      | **mAP** |
| **E-DETR** | 10.3      | 38.3    | 10.7           | 14.7    |

|            | Cityscapes |         | Corrupted Cityscapes |         |
| ---------- | ---------- | ------- | -------------------- | ------- |
| **Model**  | **D-ECE**  | **mAP** | **D-ECE**            | **mAP** |
| **E-DETR** | 9.6        | 30.5    | 9.9                  | 12.6    |

|                        | Sim10K    |         | BDD100K   |         |
| ---------------------- | --------- | ------- | --------- | ------- |
| **Model**              | **D-ECE** | **mAP** | **D-ECE** | **mAP** |
| **BPC**                | 6.1       | 65.4    | 6.3       | 23.4    |
| **Cal-DETR**           | 6.2       | 65.9    | 6.3       | 23.8    |
| **E-DETR [12 Epochs]** | 7.4       | 61.3    | 7.2       | 20.6    |


We achieved state-of-the-art results on the KITTI benchmark, surpassing our baseline model, EVCenterNet (Nallapareddy et al. (2023) IROS), by **7.7 percentage points** when averaged across all categories. Additionally, we reduced our error by **1.8 points** on the D-ECE metric and outperformed EVCenterNet across all our out-of-domain datasets.

We beat our baselines Cal-DETR (Munir, et al. (2024) NeurIPS) and BPC (Munir, et al. (2023) CVPR)
by **7.3 points mAP** and **8.9 points mAP** on in-domain Cityscapes, respectively. And achieve a comparable error for D-ECE.

And by **9.3 points mAP** and **9.2 points mAP** on out-domain Foggy Cityscapes. And **7.2 mAP** and **7.3 mAP** on out-domain BDD100K, respectively.


In our updated revision, we have revised the network architecture to better illustrate the learning of distributions. The methodology section has also been improved for clarification from line 230. We believe that the contributions outlined in the introduction, methodology, and ablation studies are now more cohesively linked.

---

### Note · Authors · 2025-01-31

I have read and agree with the venue's withdrawal policy on behalf of myself and my co-authors.